Optimizing chlorine dioxide treatment for enhanced post-harvest storage quality of Toona Sinensis

Zeng Rui 1 2 3 zengrui829@126.com
Gao Yunhong 4
Zheng Mingmin 1 2
Lai Min 2 3
JiGu Yishi 2
Chen Jiayu 2
Pei Yun 2
Farooq Muhammad Umer 5
Al-Anazi Khalid Mashay 6
Farah Mohammad Abul 6
1 Sichuan Provincial Key Laboratory for Development and Utilization of Characteristic Horticultural Biological Resources, College of Chemistry and Life Sciences, Chengdu Normal University , Chengdu, Sichuan , China
2 College of Chemistry and Life Science, Chengdu Normal University , Chengdu, Sichuan , China
3 Institute of Food Fermentation, Chengdu Normal University , Chengdu, Sichuan , China
4 Sichuan Yizhong Agricultural Development Company , Chengdu, Sichuan , China
5 Wheat Research Institute, Agriculture Department, Ayub Agricultural Research Institute , Faisalabad, Punjab , Pakistan
6 Department of Zoology, College of Science, King Saud University , Riyadh , Saudi Arabia
Marunaka Yoshinori
Electronic publication date: 2024 Nov 5
Publication date: 2024
Volume: 12
Electronic Location ID: e18346
Received 2024 Jun 21; Accepted 2024 Sep 27
Copyright: © 2024 Zeng et al.
Copyright year: 2024
Copyright holder: Zeng et al.
License: This is an open access article distributed under the terms of the Creative Commons Attribution License, which permits unrestricted use, distribution, reproduction and adaptation in any medium and for any purpose provided that it is properly attributed. For attribution, the original author(s), title, publication source (PeerJ) and either DOI or URL of the article must be cited.
License URL: https://creativecommons.org/licenses/by/4.0/

Keywords: Toona sinensis, Chlorine dioxide, Preservation, Storage, Quality, Flavonoid, Vitamin C

Funding: High-Level Talent Introduction Program of Chengdu Normal University YJRC2021-03 Teaching Reform Project of Chengdu Normal University 2023JG19 College Students’ Innovation and Entrepreneurship Project S202114389165, S202214389166 Sichuan Provincial Key Laboratory of Philosophy and Social Sciences for Monitoring and Evaluation of Rural Land Utilization NDZDSB2023004 Researchers Supporting Project number RSPD2024R694 King Saud University This research was funded by the High-Level Talent Introduction Program of Chengdu Normal University (YJRC2021-03), the Teaching Reform Project of Chengdu Normal University (2023JG19), the College Students’ Innovation and Entrepreneurship Project (S202114389165, S202214389166), the Sichuan Provincial Key Laboratory of Philosophy and Social Sciences for Monitoring and Evaluation of Rural Land Utilization (NDZDSB2023004) and by the Researchers Supporting Project number (RSPD2024R694), King Saud University, Riyadh, Saudi Arabia. The funders had no role in study design, data collection and analysis, decision to publish, or preparation of the manuscript.

==============================
This study investigated the impact of chlorine dioxide (ClO2) treatment on the storage quality of Toona sinensis after harvesting. Toona sinensis samples treated with different concentrations (0.2, 0.4, 0.8, 1.6 mg/L) of chlorine dioxide and stored at (4 ± 1) °C, with sampling test every 2nd day. The changes in sensory, nutritional, and chlorine dioxide residues of T. sinensis were checked regularly. Results revealed that moderate (0.4~0.8 mg/L) chlorine dioxide concentrations maintained sensory quality, inhibited T. sinensis weight loss, slowed flavonoid and vitamin C content depletion, reduced nitrite content, and enhanced superoxide dismutase activity. The chlorine dioxide resides in T. sinensis were below the national standards (<2.0 mg/kg; GB 5009.244-2016). Overall, 0.4~0.8 mg/L chlorine dioxide treatment showed optimal effects on T. sinensis, providing a scientific basis for extended storage and preservation of T. sinensis.

Introduction

Toona sinensis (Toon), commonly known as Chinese Toon or Chinese mahogany, is indeed an interesting plant to consider for sustainability, it belongs to Meliaceae family (Peng et al., 2019). It has cultural significance in Chinese cuisine and traditional medicine. The plant parts i.e., roots, stems, leaves, and fruits are valued for their unique flavor and nutritional benefits (Wang, Yang & Zhang, 2007). The shoots are used as vegetables (Liao et al., 2007). Additionally, different parts of the tree, such as the mature leaves, bark and seeds, have been used in traditional medicine for various purposes (Guanying et al., 2019). The leaves are pinnate, compound, and alternate, consisting of multiple small leaflets, usually buds are commercialized in spring. The immature leaves are reddish-purple color, with a unique taste and a special aroma (Jiang et al., 2020; Peng et al., 2019). Toon extracts and chemical components have good antibacterial, anti-inflammatory, analgesic, antioxidant, neuroprotective, anti-cancer, hypoglycemic and other biological activities (Chanyuan et al., 2020). T. sinensis is beloved for its robust aroma and exceptional nutritional content, however, there are challenges in post-harvest preservation, due to seasonal variations. The delicacy of fresh Toon leaves makes them susceptible to water loss, wilting, leaf drop, and mold during transportation and storage (Wang et al., 2024b; Xu et al., 2022). These issues not only hinder direct sales and further processing but also limit the economic potential of T. sinensis, diminishing its edibility and safety. Addressing these challenges is crucial for unlocking the full value of this esteemed ingredient and ensuring its availability and quality for consumers (Xin et al., 2019).

Chlorine dioxide (ClO2) is globally recognized as a potent disinfectant, classified as an A1 disinfectant due to its strong oxidizing and bactericidal properties (Mengqiao et al., 2021). Unlike many other sterilization agents, ClO2 leaves no harmful residues or odors during the sterilization process, ensuring food safety without compromising flavor or sensory quality (Zhang et al., 2023a). It has been successfully employed in preserving various fruits and vegetables, effectively inhibiting the growth of spoilage bacteria without reacting with fatty acids (Wang et al., 2024a; Zhang et al., 2023b). This preservation method keeps food fresh, minimizes quality loss, preserves appearance, and slows the degradation of essential nutrients like soluble solids, titratable acid, vitamin C (Vc), and chlorophyll. Additionally, it enhances the activity of peroxidase (POD) while inhibiting polyphenol oxidase (PPO), further reducing decay in produce (fruits/vegetables) (Chao et al., 2020; Hui et al., 2020; Shengzhi et al., 2020; Xingqiang et al., 2020; Yunfei et al., 2020).

Despite the economic potential of T. sinensis, its value is limited due to concentrated harvesting periods, short sales windows, and post-harvest storage challenges. Current preservation techniques for T. sinensis encompass physical (Yumeng et al., 2023), chemical (Liqiong et al., 2019; Shaohua et al., 2019), biological (Xihang, 2022) and combined methods (Shaohua et al., 2021; Yongqing et al., 2014). In this study, we investigated the effects of varying chlorine dioxide concentrations on various physiological parameters during T. sinensis post-harvest storage. Our goal was to identify an effective method to enhance post-harvest quality and prolong shelf life, thereby contributing to the advancement of the T. sinensis industry and meeting consumer demands.

Materials and Methods

Material collection

T. sinensis was obtained from the Toona base in Guangming Village, Yaodu Town, Qingbaijiang District, Chengdu City, Sichuan Province. These Chinese Toons are all clones and cultivated. Fresh Toona leaves of uniform size, free from pests and diseases, and with intact appearance were chosen as raw materials and transported to the laboratory of Chengdu Normal University within 1 h of harvest for further analysis. The length of each selected Toon shoot was 5–10 cm.

The following instruments were utilized in study: Ultraviolet-visible spectrophotometer 752G (Shanghai Jingqi Instrument Co., Ltd., Shanghai, China), ultra-low temperature refrigerator DW-25L300 (AUCMA Co., Ltd., China), desktop refrigerated centrifuge 5810R (Eppendorf, Germany), large-capacity low-speed centrifuge TDL-5A (Jiangsu Jinyi Instrument Technology Co., Ltd., Changzhou, Jiangsu, China), ultrasonic cleaner KH3200B (Kunshan Hechuang Ultrasonic Instrument Co., Ltd., Shanghai, China), analytical electronic balance FA1004 (Shanghai Liangping Instrumentation Co., Ltd., uyao, Shanghai, China).

Preparation of ClO2 solution

The solid ClO2 powder was poured in distilled water and stored at room temperature away from light for 3 days to allow the solid ClO2 powder to completely dissolve. It was stored at 4 °C for further studies. The concentration of the ClO2 solution was determined according to the national standard for stable chlorine dioxide solution (GB/T 20783-2006) (National Health and Family Planning Commission of China, 2016a).

Samples pretreatment

The selected Toon samples were randomly assigned five chlorine dioxide (ClO2) treatment groups (as shown in Table 1), with each group triplicated. Each sample weighing 100g was placed in a perforated ziploc bag made of polyethylene (PE). The inner walls of the ziploc bags were sprayed with a ClO2 solution evenly using a sprayer. The bags were sealed and stored at 20 ± 2 °C for 2, 4, 6, 8 and 10 days. The permissible standard of ClO2 in food products is followed, which refers to the GB 5009.244-2016 (“National Food Safety Standard of the People’s Republic of China Determination of Chlorine Dioxide in Food”), requires that the surface residue of ClO2 treated fruits and vegetables should not exceed 2 mg/kg (National Health and Family Planning Commission of China, 2016b).

Table 1 The ClO2 treatment groups used in the study.

Treatment groups	ClO2 concentration mg/L	
CK	0	
S1	0.2	
S2	0.4	
S3	0.8	
S4	1.6	

Determination of weight loss rate

The weight loss rate was determined as.

(1) WeightLossRate=Toonqualitybeforestorage−ToonqualityafterstorageToonqualitybeforestorage.

Determination of sensory index

The browning and decaying are important indicators that affect the sensory perception of Chinese Toon. The degree of browning may be related to the decomposition rate of chlorophyll. Changes in color and smell are the most intuitive manifestations of quality changes during storage. The comprehensive scoring method assesses the appearance, morphology, texture, and distinctive odor of T. sinensis as evaluation criteria. According to the guidelines outlined in Table 2, a score of >20 points is allocated to each criterion, resulting in a total score of 100 points. We selected twenty people to evaluate the Chinese Toon, ensuring that the evaluation personnel are defined each time, and take the average of the scoring from the same people. Evaluators calculate the average sensory evaluation score for each group of T. sinensis based on actual observations. If the sensory score falls below 60 points, it says that the T. sinensis has lost its commercial value. Sensory evaluation was conducted in a professional sensory evaluation room.

Table 2 Sensory scale scoring on freshness evaluation of T. sinensis by evaluators.

Sensory index	Scoring	Grading criteria	
Appearance	20	a. The color is bright, and the surface has an oily texture (16~20 points).	
b. The color is bright, and the surface has a slight oily texture (11~15 points).	
c. The appearance is average, and the surface has no oily surface texture (6~10 points).	
d. Browning occurs, no commercial value (0~5 points).	
Morphology	30	a. Good morphology, smooth leaf edges, no aging phenomenon (26~30 points).	
b. Good morphology, flat leaf edge, no aging phenomenon (20~25 points).	
c. Leaf micro-curl, a small amount of aging phenomenon (11~19 points).	
d. The leaf curl and aging phenomenon is serious (0~10 points).	
Texture	25	a. The stems and leaves are straight, and there is no wilting and decay (21~25 points).	
b. The stems and leaves are straight, and there is no wilting and decay (13~20 min).	
c. It rarely wilts and decays, and the leaf edge is a small amount of dry brown (6~12 points).	
d. Most of them wilted and rotted, and the tissues were dry brown (0~5 points).	
Distinctive odor	25	a. Rich (21~25 points).	
b. Stronger (13~20 points).	
c. Light (6~12 points).	
d. Light, with a rotten smell (0~5 points).	

Determination of total flavonoid content

For the determination of total flavonoid content, the aluminum nitrate method followed by Chun-Cui et al. (2024) was utilized.

1) Weigh 0.50 g of crushed T. sinensis into a centrifuge tube, then add 10 mL of 50% ethanol. Sonicate the mixture at a temperature of 60 °C for 1 h. Place the centrifuge tube into a centrifuge and spin it at a speed of 5,000 r/min for 20 min. Extract 1 mL of supernatant and transfer it to a colorimetric tube with a stopper. Initially, add 0.3 mL of 5% sodium nitrite (NaNO2) solution and mix thoroughly. Allow it to stand for 6 min. Then, add 0.3 mL of 10% aluminum nitrate (Al (NO3)3) solution and mix well, let it stand for another 6 min. Add 4 mL of 4% sodium hydroxide (NaOH) solution and adjust the volume with 50% ethanol to 10 mL and shake thoroughly. After standing for 15 min, measure the absorbance at a wavelength of 510 nm without adding the extract. Each treatment was triplicated, and absorbance was estimated three times per test.

2) To prepare the standard curve, dissolve 10 mg of rutin standard sample in 50% ethanol and transfer it to a 50 mL volumetric flask. Dilute it to obtain a 0.2 mg/mL rutin standard solution. Then, take aliquots of 0.0, 0.5, 1.0, 1.5, 2.0, and 2.5 mL from the standard solution and transfer each to a 10 mL volumetric flask. Add NaNO2 solution, Al (NO3)3 solution, and NaOH solution to each flask, and adjust the volume with ethanol as refined earlier. Shake the solutions thoroughly and allow them to stand for 15 min. Use the solution with 0 mL of the rutin standard as the reference. Measure the absorbance at 510 nm and plot a standard curve.

3) Total flavonoid calculation formula:

(2) Flavonoids(mg/g)=m×VM

where: m is the total flavonoid content in the 1.0 mL sample after ultrasonication (mg/g), V denotes the total volume of the extract (mL), and M is the weight of the T. sinensis sample (g).

Determination of vitamin C content

For the determination of vitamin C (Vc) content the national standard (GB5009.86-2016) procedure was followed (National Health and Family Planning Commission of China, 2016d).

1) Calibrating ascorbic acid solution: Accurately draw 1 mL of ascorbic acid standard solution into a 50 mL volumetric flask. Add 10 mL of oxalic acid solution, stir well, and then titrate with 2,6-dichloroindophenol solution until the solution turns pink, ensuring the color stays stable for 15 s without fading. Concurrently, another 10 mL of oxalic acid solution is utilized as a blank test.

2) Sample determination: Accurately weigh 1.0 g of crushed T. sinensis. Add a small amount of oxalic acid solution and transfer it to a 50 mL volumetric flask. Adjust the volume with oxalic acid solution, shake thoroughly and filter. Add white clay to the filter and repeat the filtration process. Collect the solution for later use. Pipette 10.0 mL of the filtrate into a conical flask. Titrate with the calibrated 2,6-dichloroindophenol solution until a reddish color appears and is still stable for 15 s. Note the amount of dye used.

3) Calculation formula:

(3) Titer:T(mg/mL)=C×VV1−V0

where: C stands for the mass concentration of the ascorbic acid standard solution (mg/mL), while V denotes the volume of the ascorbic acid standard solution (mL). V1 is the volume of 2,6-dichloroindophenol solution consumed during the titration of the ascorbic acid standard solution (mL), and V0 is the volume of 2,6-dichloroindophenol solution consumed during the titration blank (mL).

(4) VitaminC:Vc(mg/100g)=(Va−Vb)×T×Am×100

where: Va is the volume of 2,6-dichloroindophenol solution consumed during the titration of the sample (mL), while Vb denotes the volume of 2,6-dichloroindophenol solution consumed for the titration blank (mL). T signifies the titer of the 2,6-dichloroindophenol solution (mg/mL), A is the dilution factor, and m shows the mass of the sample (g).

Determination of the nitrite content

For the determination of the nitrite content, the national standard (GB 5009.33-2016) procedure was followed (National Health and Family Planning Commission of China, 2016c).

1) Sample determination: Accurately weigh 2.5 g of crushed T. sinensis. Add 6.25 mL of saturated borax solution, in a 50 mL of water at 70 °C. Mix thoroughly and heat the mixture in a boiling water bath for 15 min. Remove from heat and allow to cool to room temperature. Transfer the above extract quantitatively to a 100 mL volumetric flask, adding 2.5 mL of potassium ferrocyanide solution dropwise while rotating the flask. Shake well and then add zinc acetate solution. Adjust the volume to the mark with water, shake well, and let it stand for another 30 min before filtering. After filtration, aspirate 20.0 mL of the filtrate into a 25 mL colorimetric tube. Add 1 mL of p-aminobenzene sulfur solution and mix well. After standing for 3–5 min, add 0.5 mL of naphthalene ethylenediamine hydrochloride solution. Adjust the volume to the mark with water, mix thoroughly and allow it to stand for 15 min.

2) Drawing the standard curve: Take 0, 2, 4, 6, 8, 12, 20, and 40 mL of the sodium nitrite standard solution and place them in 8 (50 mL) colorimetric tubes with stoppers. Add 2 mL of p-aminobenzene sulfurous solution to each tube and mix well. After standing for 3–5 min, add 1 mL of naphthalene hydrochloride ethylenediamine solution to each tube. Adjust the volume to the mark with water, mix well, and let it stand for 15 min. Use a 1 cm cuvette and adjust the zero point with a tube having 0 mL of the standard solution. Measure the absorbance value at a wavelength of 538 nm and draw the standard curve.

3) Calculation formula:

(5) X=m2×1000m3×(V1V0)×1000

where: X is the nitrite content in the sample (mg/kg), m2 is the mass of sodium nitrite in the sample solution (μg), m3 is the mass of the sample (g), V1 is the volume of the sample for determination (mL), V0 is the total volume of the sample treatment solution (mL).

Determination of superoxide dismutase activity (SOD)

The SOD activity was determined by nitroblue tetrazolium (NBT) colorimetry method as suggested earlier (Jiankang, Weibo & Yumei, 2007). The NBT colorimetric method quantifies SOD activity by measuring the inhibition of the reduction of NBT to formazan. The less formazan formed, the higher the SOD activity in the sample. The decrease in color intensity (blue color) is measured using a spectrophotometer at a specific wavelength (usually around 560 nm). The SOD activity is calculated based on the inhibition of NBT reduction.

Determination of chlorine dioxide residue

The residual ClO2 accumulation was tested using the DPD ultraviolet spectrophotometric method of “National Food Safety Standard Determination of Chlorine Dioxide in Food (2016)” (GB 5009.244-2016), which requires that the surface residue of fruits and vegetables treated with ClO2 should not exceed 2 mg/kg (National Health and Family Planning Commission of China, 2016b).

Data analysis

Three different T. sinensis samples were selected in triplicate. Five chlorine dioxide (ClO2) treatment groups were used including control and four levels. The analysis of variance followed randomized complete block design was used. For the data calculation, plotting, origin (2018) software was used (OriginLab, Northampton, MA, USA). The data processing system (DPS; DriverPacks Base) software was used for statistical analysis of difference significance (ANOVA) and then Duncan’s new multiple range method test was performed, followed by Tukey’s statistical analysis to find significant differences between groups at 5% level of significance (P < 0.05).

Results

Weight loss is a key indicator of storage quality in Toon. When Toon sprouts are kept in the open air, various enzymes accelerate the conversion of sugars into alanine. This alanine then enters a series of tricarboxylic acid cycles, generating heat that leads to water loss. Studies have shown that water and dry matter loss during storage period of Toon is affected, and water loss and transpiration is a key factor (Baigetumur et al., 2021). The weight loss seen in the Toon from the five treatment groups (Fig. 1) showed an upward trend as storage time increased. The CK group showed the highest weight loss, reaching 5.58% by the 8th day, significantly higher than the other treatment groups (P < 0.01). In contrast, the S3 group had the lowest weight loss, significantly lower than the S1 and S4 groups (P < 0.05), but not significantly different from the S2 group (P > 0.05). By the 10th day, the S3 and S2 groups had weight losses of 5.40% and 6.27%, respectively, both significantly lower than the other three groups (P < 0.01). The results say that the concentrations in the S2 and S3 groups could not effectively reduce weight loss and maintain the freshness of Toon. However, chlorine dioxide slows down the respiration and delays the wilting of the Toon.

Figure 1 Effect of different chlorine dioxide (ClO2) treatments on weight loss of T. sinensis.

As storage time increased, the appearance of both the CK and experimental groups deteriorated (Fig. 2), with the sensory quality of the Toon declining, including loss of color and luster, leaf curling, and eventual wilting. Based on the sensory scores (Fig. 2A), there was no significant difference among the treatment groups during the first 6 days. However, by day 8, the sensory scores for the CK and S4 groups were significantly lower than those for the S1, S2 and S3 groups (P < 0.01). The differences between the S1, S2 and S3 groups were not significant (P > 0.05) on day 8. On the 10th day, the scores for the S2 and S3 groups were not significantly different (P > 0.05), but they were significantly higher than those for the S1, S4 and CK groups (P < 0.01). These results show that the S2 and S3 chlorine dioxide treatment groups could delay the sensory quality deterioration of Toon over time. Between days 2 and 8, the degree of leaf wilting in the chlorine dioxide-treated Toon buds was significantly lower than in the control group. While chlorine dioxide could not completely inhibit wilting, it significantly slowed the decline in quality of fresh Toon buds in the short term.

Figure 2 Effect of different chlorine dioxide treatments on sensory evaluation of T. sinensis.

(A) The sensory score of different chlorine dioxide treatments on Toons; (B) the appearance of Toon due to different chlorine dioxide treatments.

A decreasing trend reside in total flavonoid activity was seen across the different treatment groups as storage time increased (Fig. 3). This trend may be related to the intensified respiration of Toon after harvesting. The S2 group showed the slowest decline in flavonoids, with contents range from 9.20, 8.81, and 8.44 mg/g on the 4th, 6th, and 8th days, respectively, which were significantly higher than those in the other treatment groups (P < 0.05). By the 10th day of storage, the CK group had significantly low flavonoid content (P < 0.01), showing that chlorine dioxide can slow the depletion of total flavonoid content, with the S2 group showing the best preservation effect.

Figure 3 Effects of different chlorine dioxide treatments on the total flavonoid content of T. sinensis.

Vc is an important indicator of freshness in fruits and vegetables due to its ability to effectively remove reactive oxygen species (ROS) (Shuangfang et al., 2020). During the storage period, the vitamin C content in the CK group and other treatment groups significantly decreased from day 0 to day 10, with varying levels of retention in each group (Fig. 4). The CK group had the lowest vitamin C content, which dropped from 42.20 (mg/100g) to 3.94 (mg/100g) by the 10th day, significantly lower than in the S2 and S3 groups (P < 0.05). The S2 and S3 groups kept the highest vitamin C levels by the 10th day, with a slower decline compared to the CK group, indicating that the chlorine dioxide treatment can delay vitamin C degradation, with the S2 group showing the best preservation effect.

Figure 4 Effect of different chlorine dioxide treatments on the vitamin C content of T. sinensis.

Nitrite levels in food, including Chinese Toon (T. sinensis), are crucial for ensuring quality and safety. Nitrites, commonly found in processed meats and preserved foods, pose health risks such as dizziness, poisoning, and even cancer (ShengZhen & ShengZhen, 2013). Therefore, monitoring and understanding the dynamic changes in nitrite content during storage is essential to keeping the quality of Toon. Changes in nitrite content during storage under different treatments (Fig. 5), showed that the nitrite levels in all experimental groups were extremely lower than in the CK group (P < 0.01), though differences among the experimental groups were not significant (P > 0.05). On the 4th day, chlorine dioxide effectively inhibits nitrite increase. As storage time extended, the S2 and S3 groups had the best inhibitory effect on nitrite, keeping levels within the national food safety standards (≤40 mg/kg), the reason may be that during the storage process of Toona sinensis, the microbial activity gradually increases, ultimately increasing the nitrite contents (Xingyu, Leilei & Fengmei, 2019). The CK and S1 groups showed significant nitrite peaks on the 8th day, at 186.05 and 51.56 mg/kg respectively, both exceeding the national food safety standards (≤40 mg/kg). Chlorine dioxide concentration of 0.4~0.8 mg/L significantly reduced nitrite content.

Figure 5 Effect of different chlorine dioxide treatments on nitrite concentration of T. sinensis.

The perishability and decay of fruits and vegetables are closely related to changes in enzymatic activity (Kader, 2002). After harvesting, the metabolic activity of fruits and vegetables produces reactive oxygen species (ROS) (Hodges et al., 1999), leading to cell membrane damage and aging (Mittler, 2017). The enzyme SOD can significantly reduce the toxic effects of superoxide anions on cells and regulate the body’s oxidative metabolism (Alscher, Erturk & Heath, 2002). Results showed that chlorine dioxide concentration of 0.4 and 0.8 mg/L increased SOD enzyme activity, particularly on the 8th and 10th days of storage (Fig. 6). On the 8th day, SOD levels were 1288.57, 1209.61, and 837.19 U/g, in the ClO2 groups (0.8, 0.4 and 1.6 mg/L respectively), significantly higher than in the CK and 0.2 mg/L ClO2 groups (P < 0.01). No significant differences were seen on the 2nd, 4th, and 6th days (Fig. 6).

Figure 6 Effects of different chlorine dioxide treatments on SOD enzyme content of T. sinensis.

Apart from the beneficial impact of ClO2 on enzymes and perishability activities, the residual accumulation of chlorine dioxide must be assessed to ensure safe food supply to consumer market. Chlorine dioxide quickly decomposes into highly active chlorite and Cl-, which can easily combine with organic matter to form harmful byproducts. These byproducts can accumulate in the human body and disturbs the body’s metabolism, posing serious health risks (Abdel-Rahman, Couri & Bull, 1979). Tests showed that the residual accumulation of chlorine dioxide in Toon decreased gradually with longer storage times (Fig. 7). By the 8th day, the chlorine dioxide residues in the S1 and S2 groups were below 1.75 mg/kg, lower than the national (GB 5009.244-2016) chlorine dioxide residue standard of 2 mg/kg (National Health and Family Planning Commission of China, 2016b). On the 10th day, the chlorine dioxide residues in the S2 and S3 groups remained below the national standard, with residues of 1.60 and 1.65 mg/kg, respectively.

Figure 7 Chlorine dioxide residues in T. sinensis under different chlorine dioxide treatments.

Discussion

Chlorine dioxide is an important disinfectant categorized as A1 and is well known for its adaptive features. Apart from its beneficial aspects, a detailed study necessitates assessing its impact on various biochemical and residual effects of T. sinensis. The weight loss, sensory evaluation, quality gradient, total flavonoid, vitamin C, nitrite concentration, SOD activity and residual accumulation in Toon samples were determined to provide a complete detailed study for long term preservation and post-harvest storage of quality products. This study presents compelling evidence on the effectiveness of chlorine dioxide (ClO₂) treatments in supporting the quality and extending the shelf life of T. sinensis during storage. The post-harvest ClO2 treatment can slow down the respiration of Toon and effectively reduce the weight loss (Fengyuan et al., 2017). The sensory analysis further supports the efficacy of ClO2 in preserving the Toon freshness. The sensory quality index under different ClO2 treatment levels (Fig. 2B) disclosed that chlorine dioxide can delay visual and textural degradation (Karabulut et al., 2009). It can effectively inhibit the decay of Toon and keep good sensory quality (average around 80 points). One of the possible reason is the chlorine dioxide’s ability to destroy the stability of bacterial cell membranes, inhibit the synthesis of microbial proteins, effectively destroy microorganisms, improve the antioxidant capacity and disease resistance of Toon, and reduce the consumption of nutrients (Xiaoyu, 2006).

The results also show that chlorine dioxide can effectively slow the depletion of bioactive compounds such as total flavonoid content and vitamin C. Appropriately, the chlorine dioxide treatment of 0.4 mg/L had the high impact for retention of flavonoids and vitamin C in Toon by the day 10. Flavonoids have biological activities such as anti-cancer, antioxidant, anti-aging, anti-inflammatory, and immunity-boosting. Vitamin C is naturally reducible in products (Aldhaher, 2016; Peng et al., 2019). Fang, Zhao & Warner (2012) found that chlorine dioxide has strong oxidizing property, and its reducing effects on various compounds, including vitamin C, during the treatment of fresh produce. The results show that the balanced concentration of chlorine dioxide, appropriate treatment duration and suitable storage conditions can effectively delay the degradation of vitamin C by controlling microbial growth and reducing oxidative stress on the produce (Fang, Zhao & Warner, 2012).

Nitrite can naturally reside in food or can be added as a preservative, and its concentration is regulated due to its potential health effects. The chlorine dioxide had a significant inhibitory effect on the nitrite content especially on the 4th day of treatment application. As storage time increases, a significant nitrous peak reduction appeared on the 8th and 10th day, indicating ClO2 inhibitory effect. This difference in effectiveness ascribes the strong oxidizing properties of chlorine dioxide, which can lead to the destruction of the cell structure of T. sinensis at higher concentrations. The treatment (0.4 and 0.8 mg/L) effectively kept the quality of T. sinensis. The chlorine dioxide high concentration during the initial stages might be a reason, which inhibits the increase of nitrite content. The ClO2 treatment can help mitigate this risk by reducing nitrite levels while still providing effective antimicrobial action and subsequent nitrite formation as indicated by Zhong & Zhang (2013). The reduction of nitrite content through ClO2 treatment can be helpful in food preservation as it can inhibit the growth of nitrite-producing bacteria, thus preventing the accumulation of harmful levels of nitrites in stored foods. Especially in products where controlling nitrite levels is important for safety and regulatory compliance.

The degradation of enzymatic activity results in depletion of perishability in food products. Superoxide dismutase (SOD) is the main enzyme that scavenges ROS and plays a critical role in protecting cells from oxidative damage. The level of ROS scavenging enzymes is closely related to the storage stability and quality attributes in Chinese Toon (Lin et al., 2020; Lin et al., 2017). The SOD activity on the 8th day for 0.8 and 0.4 mg/L ClO2 treatment group was significantly higher. The changes on the 2nd, 4th, and 6th day were non-significant. The 0.4 and 0.8 mg/L concentration of chlorine dioxide improves the activity of SOD enzymes and helps in alleviating the perishability of T. sinensis. The increased SOD activity suggests that ClO2 can bolster the antioxidant defense of the Toon, thereby extending its shelf life. This is consistent with the findings of karabulut, who reported that ClO2 treatments could enhance the antioxidant enzyme activities in fresh produce, contributing to reduced oxidative stress and prolonged freshness (Karabulut et al., 2009). The improvement in SOD activity of leaves due to ClO2 treatment was also observed by Qiwei et al. (2015).

However, the potential for ClO2 residue accumulation must be carefully checked to ensure consumer safety. The findings of the current experiment show that ClO2 residue in the S2 and S3 groups were within safe limit by the 10th day of storage, suggesting its safe and effective use. This is crucial as ClO2 can decompose into by-products that, if accumulated in significant quantities, could pose health risks (Abdel-Rahman, Couri & Bull, 1979). The high concentration of chlorine dioxide residues in plant tissues is attributed to several factors (Li & Huang, 2008). Some residues may remain on the surface of the leaves or within the plant tissue, can accumulate if not adequately removed during subsequent processing/washing steps (Gómez-López, A & Gil, 2009). Therefore, proper dosage, optimal treatment duration, effective rinsing/washing and monitoring of samples should be an important practice to ensure effective reduction of harmful components in Toon without compromising its product quality or safety.

Conclusions

The results of this study indicated that the chlorine dioxide treatment at concentrations of 0.4 and 0.8 mg/L were better compared to concentrations of 1.6 and 0.2 mg/L. Key findings indicate that different ClO₂ treatments significantly affect the sensory attributes of T. sinensis, with implications for improving its storage quality. It can effectively mitigate weight loss, delay sensory deterioration, and preserve key nutritional components such as flavonoids and vitamin C for more than 8 days. Additionally, ClO2 was effective in reducing nitrite accumulation, and enhancing the activity of superoxide dismutase, which is critical for food safety. The study adhered to the national standards for chlorine dioxide residue (<2.0 mg/kg). Overall, the study demonstrates that appropriate chlorine dioxide concentration of 0.4 and 0.8 mg/L is an effective treatment for preserving the quality and extending the shelf life (for 8 days) of T. sinensis under the conditions of 20 ± 2 °C, improving the post-harvest storage.

Supplemental Information

Supplemental Information 1 Experimental data.

Additional Information and Declarations

Competing Interests

Author Contributions

Data Availability

The authors declare that they have no competing interests. Yunhong Gao is the legal representative of the Sichuan Yizhong Agricultural Development Company.

Rui Zeng conceived and designed the experiments, performed the experiments, analyzed the data, prepared figures and/or tables, authored or reviewed drafts of the article, give the funding acquisition, and approved the final draft.

Yunhong Gao performed the experiments, authored or reviewed drafts of the article, and approved the final draft.

Mingmin Zheng performed the experiments, authored or reviewed drafts of the article, and approved the final draft.

Min Lai performed the experiments, analyzed the data, authored or reviewed drafts of the article, and approved the final draft.

Yishi JiGu analyzed the data, authored or reviewed drafts of the article, and approved the final draft.

Jiayu Chen performed the experiments, analyzed the data, prepared figures and/or tables, and approved the final draft.

Yun Pei performed the experiments, authored or reviewed drafts of the article, and approved the final draft.

Muhammad Umer Farooq analyzed the data, prepared figures and/or tables, authored or reviewed drafts of the article, and approved the final draft.

Khalid Mashay Al-Anazi analyzed the data, authored or reviewed drafts of the article, give the funding acquisition, and approved the final draft.

Mohammad Abul Farah analyzed the data, authored or reviewed drafts of the article, give the funding acquisition, and approved the final draft.

The following information was supplied regarding data availability:

The raw data are available in the Supplemental File.

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
