# Peer review of "Optimizing chlorine dioxide treatment for enhanced post-harvest storage quality of Toona Sinensis"

_PeerJ, doi:10.7717/peerj.18346_

## Round 0.1 · original submission · Major Revisions

If you feel you can revise your manuscript according to the reviewers' comments, please revise your manuscript and submit it. Please also send us your written responses to each of the reviewers' comments.

Yours,

Yoshi

Prof. Yoshinori Marunaka, M.D., Ph.D.

Reviewer 1 ·

Basic reporting

Some areas of clarity required. Many sentences in the Introduction require supporting references. Review the formatting of the references. It identified the chlorine dioxide concentration requited for storage of Toona sinensis.

Abstract:
Include the analysis of data. The results are generalized. The results need to include the treatments effect of chlorine dioxide concentration by storage days on the quality parameters measured.
Line 18 Indicate the specific regular interval…….The changes in sensory, nutritional, and chlorine dioxide residues of Toona sinensis were estimated regularly.
Line 19 - Results revealed that moderate (explain the concentration) chlorine dioxide concentrations,
Lines 21 and 22 The chlorine dioxide resides (should be residues) in Toona sinensis below the national standards (indicate the standard)
Line 25 Include key words other than in the title e.g. flavonoid, Vitamin C

Introduction
Many statements lack references.
Lines 30-31-The shoots are used as vegetables (include reference/s) .
Lines 32-35- The leaves are pinnate, compound, and alternate, consisting of numerous small leaflets, usually buds are marketed in spring. When young, the leaves have a reddish-purple color, with a unique taste and a special aroma.(include reference/s).
Lines 37-40-Toona sinensis is beloved for its robust aroma and exceptional nutritional content, faces challenges in postharvest preservation, largely due to seasonal variations. The delicacy of fresh Toona leaves makes them susceptible to water loss, wilting, leaf drop, and mold during transportation and storage. Include reference/s
Line 44 (ClO2) rather than (ClO2)
Lines 45-49 Unlike many other sterilizing agents……It has been successfully employed in preserving various fruits and vegetables, effectively inhibiting the growth of spoilage bacteria…..(include references).
Line 55-Despite the economic potential of Toona sinensis (italicize)

Experimental design

Experimental design Many methods lack references. The detail conduct of the sensory evaluation needs to be described . Include and experimental design as the treatment types and quality variables. The effects of the interaction of chlorine concentrations by storage days should have been included in the analyses to identify significance. The sensory analysis methodology was not adequately described.

Materials and Methods
Many methods lack references. The detail conduct of the sensory evaluation needs to be described . Include and experimental design as the treatment types and quality attributes. The effects of the interaction of chlorine concentrations by storage days should have been included in the analyses to identify those which were significant. The sensory analysis methodology was not adequately described. Experimental design section should have been included showing the number of treatments and variables.
Lines 68-69 -Fresh Toona leaves of uniform size ( include size dimension). Is there a specific maturity level for the leaves?
Line 86- Determination of Sensory Index- Is there a reference method. How many panellists? Where was the sensory evaluation conducted Was there any sensory training?
Line 93- 93 -Determination of Total Flavonoid Content- Include reference for method.
Line 116- Vitamin C content- Include reference for the method.
Line 133 -Should the spelling be 2,6-dichloroindophenol? Please double check.
Line 141 Determination of Nitrite Content. Include reference method.
Line 165 -Determination of Superoxide Dismutase Activity (SOD).Include reference method
Line 166 The SOD activity was determined by NBT (spell out at first)
Line 170 – Include under the list of references -National Food Safety Standard Determination of Chlorine Dioxide in Food (include year)
Line 173- For data calculation and plotting origin (2018)(include reference) was used. DPS (spell out) software.
Line 172 Include an experimental design with data analysis i.e. treatments and variables.

Validity of the findings

Results
The results are displayed as Figures. Tables may have been better to show the level of significance and the separation of the mean by Duncan’s. Also, the interaction of chlorine dioxide concentration by storage days on the quality attributes of Toon sinensis should be presented. The sensory results are not clearly analyzed and interpreted.
Lines 180-181 Studies (provide the references for these studies) have shown that the loss of water and dry matter during the storage period of toon (Toon) is…. affected
Caption of Figures should be at the bottom, Need to indicate the level of significance for each Figure
Lines 190-192 - With the increase of storage time, the appearance of the CK group and the experimental group changed to different degrees (Fig. 2b). Were these changes significant on storage?

Lines 211-213- Vitamin C plays a vital role in fruits and vegetables, not only due to its effective removal of…reactive oxygen species (ROS) (include supporting reference/s)

Lines 213-214- The Vitamin C content in both the CK group and the other
214 treatment groups decreased with the increase of storage time (Fig. 4). Were these changes significant with storage time?

Lines 221-222- Nitrite may cause dizziness, poisoning or cancer to the human body--- (reference/s). Studying the dynamic changes of nitrite content in Chinese toon during storage is crucial to its quality( explain) .

Line 229 - the storage process of Toona sinensis (place in italics)
Lines 232-233- What are the appropriate chlorine dioxide concentration to significantly reduce the nitrite content?

Lines 234-238 – These sentences require supporting references.

Lines 238-239- The results showed that 0.4 and 0.8 mg/L concentration (be consistent in the manuscript write-up as in some S 1 , S2 etc were used rather than the actual concentrations) of chlorine dioxide could increase the SOD enzyme activity in toon during storage

Lines 239-240- The chlorine dioxide treatment significantly increases the SOD activity especially during 8th and 10th day of storage. Were these significant changes in other days?

Effects of chlorine dioxide and storage on the quality parameters in a Table form may have been better representation to show the differences among each treatment. By Duncan multiple range. Also, the interaction of levels of chlorine dioxide by storage effects.

Line 241-242- day of storage, SOD contents were found as 1288.57 U/g, 1209.61 U/g and 837.19 U/g, in S3, S2242 and S4 …. It (This is not clear as to what “it” refers to)…. was significantly higher than that of the CK and S1 groups (P<0.01).

Discussion
This section needs organizing and presenting the main findings followed by the discussion. Discuss in reference to the specific findings e.g. Figs, Tables, Italicise Toona sinensis throughout the manuscript e.g. line 304.
Line 254- Was the economic impact investigated?......complete phenology, impact on various biochemical, residual, and economical impact on Toona
Lines 244-245- residual accumulation of chlorine dioxide must be tested to ensure safe food supply to consumer (elaborate with references on these effects) market.
Line 261- State the specific sensory quality and present the scores to indicate good sensory quality.
Line 282 -283 Chlorine dioxide improves the activity of SOD enzymes. Is this at all levels of chlorine dioxide and helps….? See Fig 6. Clarify.
Lines 267-268- Flavonoids have biological activities such as anti-cancer, antioxidant, anti-aging, anti-inflammatory, and immunity-boosting (include reference to support.
Line 271-272 How did chlorine dioxide concentration, treatment duration and storage conditions specifically delay the degradation of Vitamin C? Any specifics?....
Lines 278-279 - …while still providing effective antimicrobial action… (a supporting reference)
Lines 281-282 - The degradation of enzymatic activity results in degradation of perishability in food products.
Lines 285-290-These sentences lack references
Lines 283-285 The results of Cai ( date and reference) were in consistent with present, where ….Check the clarity of this sentence.
Conclusion
Needs to include the findings which also show the interactions of the effects of chlorine dioxide concentration and storage.
Lines 294-295 The results of this study indicate that chlorine dioxide treatment at concentrations of 0.4 mg/L295 and 0.8 mg/L yielded better outcomes ( which specific outcomes?) compared to concentrations of 1.6 mg/L and 0.2 mg/L.
Lines 298-299- The optimal treatment of concentrations(what were these optimal concentrations for how long? ?) were efficient for preserving total flavonoids and vitamin C…
Line 204 - conditions of 20±2°C for how long.?

Additional comments

References
These were not consistent with the author guidelines, such as formatting e.g. Ganhui, Min L, Xiuli C, Yanmei T, Duc RC, Xiaohua L, Shifu C, and Mahuawei. 2020. Research on the safety evaluation of chlorine dioxide in the storage, transportation and preservation of fruits, vegetables and aquatic products. food technology e 45:36-43. 10.13684/j.cnki.spkj.2020.11.007
e.g. Food Technology
Liqiong W, Shaohua L, Cunkun C, Huijie Z, Hongxia L, and Wentao X. 2019. Effects of 3 different preservation methods on the storage quality of Chinese toon. Food research and development 40:150-155. e.g. Food Research and Development

In the data sheet – There were some values which had more than 2 digits after the decimal points e.g 1.716666667; 1.483333333

·

Basic reporting

The manuscript is easy to follow but there should be a comprehensive evaluation of the grammar and consistency with some texts (see general comments)

Experimental design

No comments

Validity of the findings

In the discussion, there are not enough references to support the results.

Additional comments

Line 22: were below
Line 28: sustainability, it belongs
Line 34: The immature leaves are reddish-purple
Line 38: content, however, there are challenges
Line 35 and 39: There should be consistency in the use of Toon and Toona
Line 55, 57, 88, 90, 91/92, 123, 143, 223, 229, 283, 297, 304: italicized Toona sinensis
Consistency in post-harvest or postharvest; ClO2 or ClO2; use one throughout the manuscript.
Line 66: Toona sinensis was sourced
Line 80: Ziploc
Line 94: For the determination of total flavonoid content, the following… similarly for lines 142, 168
Line 95: Weigh 0.50g
Lines 98- 110: the first mention of the chemicals should be identified e.g. NaNO2, A1(NO3)3, etc.
Line 100: mix well, let it stand for another 6 minutes
Line 101: with 50% ethanol to 10 mL and shake thoroughly
Line 107: How much of the NaNO2, A1(NO3)3, and NaOH was added to each flask?
Line 108: Shake the solutions thoroughly and allow them to stand for 15 minutes.
Lin3 117: For the…, the following…
Line 120, 127: Are you referring to 2,6- dichloroindophenol solution?
Line 127: Do you want a reddish or pink color?
Line 154: of the nitrite…
Line 173: for the data calculation, plotting…
Line 179: Weight loss is an important indicator to measure storage quality of toon.
Line 181: and transpiration is a key factor.
Line 184: while weight loss of the S3 group was the lowest,…
Line 190: could not effectively reduce the weight loss rate and maintain the freshness of toon.
Line 191: respiration of toon
Line 192: As storage time increase, the appearance of the CK and the experimental group changed…
Line 193: the sensory quality of toon worsen,
Line 194: , and the ultimate wilting of the plant.
Line 196: seen. However, on day 8, the sensory scores of the CK and S4 groups were significantly lower than S1, S2, and…
Line 198: were not significant (p>0.05) on day 8.
Line 203: in the different treatment group
Line 205: reside in
Line 213: measure freshness of fruits and vegetables
Consistency in the way 4th or 4th is written
Line 241: SOD contents were 1288.57 U/g…
Lin3 242: The SOD content was significantly….
Line 245: Thereby, after testing
Line 252: and is well known
Line 259: as found by Fengyuan et al. (2017).
Line 266: The chloride dioxide in the S2 group (0.4 mg/L) had…
Line 274: especially on the …
Line 275: As storage time increase, a …
Line 278, 279, 285, 290: ClO2
Line 284: where Cai and colleagues
Line 285: Cai in-text citation and reference is missing

---

## Round 0.2 · Minor Revisions

Please revise your manuscript according to the reviewer's final comments and resubmit the revised one.

Yours,
Yoshi
Prof. Yoshinori Marunaka, M.D., Ph.D

Reviewer 1 ·

Basic reporting

The revised manuscript has been satisfactory revised.
'No comment'

Experimental design

'No comment

Validity of the findings

'No comment'

Additional comments

None

·

Basic reporting

1. Line 155 mentions a pink colour whereas line 163 mentions a reddish colour. Which is it?
2. Lines 214 and 216, in text citation written twice.
Please pay attention to toon or Toon in the text.

Please go through the document to ensure that there are no grammatical errors and there are proper punctation.

Experimental design

No comment.

Validity of the findings

No comment.

---

## Round 0.3 · accepted · Accept

Congratulations, your article is now acceptable!

Yours,
Yoshi
Prof. Yoshinori Marunaka, M.D., Ph.D.